# Memantine and Riluzole Exacerbate, Rather Than Ameliorate Behavioral Deficits Induced by 8-OH-DPAT Sensitization in a Spatial Task

**DOI:** 10.3390/biom11071007

**Published:** 2021-07-09

**Authors:** Martina Janikova, Karolina Mainerova, Iveta Vojtechova, Tomas Petrasek, Jan Svoboda, Ales Stuchlik

**Affiliations:** 1Institute of Physiology of the Czech Academy of Sciences, Videnska 1083, 142 20 Prague, Czech Republic; karolinamainerova@gmail.com (K.M.); Iveta.Vojtechova@nudz.cz (I.V.); Tomas.Petrasek@nudz.cz (T.P.); jan.svoboda@fgu.cas.cz (J.S.); 2First Faculty of Medicine, Charles University, Katerinska 1660/12, 121 08 Prague, Czech Republic; 3National Institute of Mental Health, Topolova 748, 250 67 Klecany, Czech Republic

**Keywords:** obsessive-compulsive disorder, 8-OH-DPAT, memantine, riluzole, spatial learning, memory

## Abstract

Chronic sensitization to serotonin 1A and 7 receptors agonist 8-OH-DPAT induces compulsive checking and perseverative behavior. As such, it has been used to model obsessive-compulsive disorder (OCD)-like behavior in mice and rats. In this study, we tested spatial learning in the 8-OH-DPAT model of OCD and the effect of co-administration of memantine and riluzole—glutamate-modulating agents that have been shown to be effective in several clinical trials. Rats were tested in the active place avoidance task in the Carousel maze, where they learned to avoid the visually imperceptible shock sector. All rats were subcutaneously injected with 8-OH-DPAT (0.25 mg/kg) or saline (control group) during habituation. During acquisition, they were pretreated with riluzole (1 mg/kg), memantine (1 mg/kg), or saline solution 30 min before each session and injected with 8-OH-DPAT (“OH” groups) or saline (“saline” groups) right before the experiment. We found that repeated application of 8-OH-DPAT during both habituation and acquisition significantly increased locomotion, but it impaired the ability to avoid the shock sector. However, the application of 8-OH-DPAT in habituation had no impact on the learning process if discontinued in acquisition. Similarly, memantine and riluzole did not affect the measured parameters in the “saline” groups, but in the “OH” groups, they significantly increased locomotion. In addition, riluzole increased the number of entrances and decreased the maximum time avoided of the shock sector. We conclude that monotherapy with glutamate-modulating agents does not reduce but exacerbates cognitive symptoms in the animal model of OCD.

## 1. Introduction

Obsessive-compulsive disorder (OCD) is a severe neuropsychiatric disorder affecting 1–3% of the population worldwide [1]. It is often chronic and can be very debilitating. Patients with OCD suffer from obsessions, which are recurring intrusive thoughts, and compulsions, which are ritualized stereotypic behaviors usually driven by the anxiety arising from the obsessions [2]. Patients describe the mechanism of OCD as anxiety created by obsessions being released with compulsions. This cycle is often very time and resource consuming and can destroy one’s ability to concentrate or perform basic daily tasks [2]. Apart from the core symptoms of OCD—obsessions and compulsions–deficits in executive functions and other cognitive domains have been described [3,4]. Patients with OCD have decreased cognitive flexibility measured in different set-shifting tasks [4] and also worse results in spatial cognitive flexibility tasks in virtual reality [5,6,7].

Disruption of neurotransmitter systems is considered to play a role in the pathophysiology of OCD (for a review, see Goodman et al. [8]). The hypothesis of the involvement of serotonin in the pathophysiology of OCD is popular, mainly due to the effectiveness of selective serotonin reuptake inhibitors (SSRIs) in the treatment of OCD. Additionally, an association of OCD with genes coding components of the serotonin system, such as monoamine oxidase A, or genes encoding the serotonin receptors has been shown (reviewed by Derksen et al. [9]). Moreover, agonizing serotonin receptors with meta-chlorophenyl piperazine (mCPP) and antagonizing serotonin 2A and 2C receptors with ritanserin exacerbates OCD symptoms in humans, as well as in animal models [10,11]. Contrarily, agonizing serotonin 1A/2A/2C receptors with psilocybin reduces OCD symptoms [12]. The specificity of serotonin involvement in OCD is not clear yet. It is possible that other neurotransmitter systems, such as the glutamate system, are involved, and this can be underlined by the fact that approximately 10% to 60% of patients still do not fully respond to SSRIs [13]. Glutamate is the primary excitatory neurotransmitter in the brain, and its function seems disrupted in patients with OCD [14]. Higher glutamate concentrations were found in patients with OCD in the cerebrospinal fluid [15], the orbitofrontal cortex [16], and caudate nucleus [17], and lower in the anterior cingulate cortex [17]. Therefore, there have been attempts to study glutamate-modulating agents as a possible treatment of OCD. Drugs with different mechanisms of action have been tested, such as N-methyl-d-aspartate (NMDA) receptor antagonist amantadine, a partial agonist of NMDA receptor D-cycloserine, or NMDA receptor antagonist ketamine [18,19,20]. Several clinical trials and case reports suggested the efficiency of memantine (a non-competitive low-affinity NMDA receptor antagonist) and riluzole (a drug that decreases presynaptic glutamate release by blocking sodium channels). Riluzole is an anticonvulsant drug, nowadays officially used for the treatment of amyotrophic lateral sclerosis, and off-label, it is used for the treatment of some psychiatric conditions, including OCD [21,22]. Memantine is now used for the treatment of severe Alzheimer’s disease. Several case studies have reported its efficacy for patients with treatment-refractory OCD [23,24]. In a meta-analysis of double-blinded, placebo-controlled, randomized studies made by Kishi et al. [25], memantine was valued as a valid treatment for patients with SSRI-refractory OCD symptoms. Adding memantine to the SSRIs significantly improved symptoms of OCD in patients [26].

Animal models of OCD are important tools to study this disorder with yet unknown pathophysiology. They also serve for designing and validating possible treatments. One of the pharmacological animal models of OCD is the chronic sensitization to 8-OH-DPAT, an agonist of serotonin 1A and 7 receptors. Application of this drug induces behavioral sensitization, which leads to perseverative and compulsive behaviors [27,28]. Both rats and mice treated with 8-OH-DPAT exhibit perseverative behavior and decreased spontaneous alternation in Y-maze and T-maze [29,30] and repetitive stereotypical behavior in an open-field arena [27]. In this study, we aimed to test the effects of riluzole and memantine on locomotor activity and spatial learning in the 8-OH-DPAT model of OCD. We tested these domains in chronically sensitized rats after acute administration of 8-OH-DPAT and also in chronically sensitized rats, but not after acute administration of 8-OH-DPAT. We used an acquisition configuration in the active place avoidance task on a rotating arena (Carousel maze), which is a well-established task for rats and mice, allowing simultaneous assessment of spatial learning and locomotor stimulation [31,32] (for a review, see Stuchlík et al. [33]).

## 2. Materials and Methods

### 2.1. Animals

Adult (4–5 months old) male Long–Evans rats from the breeding colony of the Institute of Physiology of the Czech Academy of Sciences were used in this experiment (98 animals in total). At the beginning of the experiment, they weighed approximately 400–500 g. Rats were housed in pairs in an animal room with a stable temperature and 12/12 light/dark cycle, with food and water always freely available. The acquisition testing was preceded by 5 days of handling and 10 sessions of habituation to the experimental arena, during which sensitization to the drugs took place.

### 2.2. Drugs and Design

8-OH-DPAT (Sigma-Aldrich, Prague, Czech Republic, Cat. No. H8520), Memantine hydrochloride (Sigma-Aldrich, Prague, Czech Republic, Cat. No. M9292), and riluzole (Sigma-Aldrich, Prague, Czech Republic, Cat. No. R116) were dissolved in sterile saline (0.9% NaCl) at concentrations of 0.25 mg/mL, 1 mg/mL, and 1 mg/mL, respectively. The experiment consisted of a habituation phase and an acquisition phase. In the habituation phase, subcutaneous injection of 8-OH-DPAT (N = 84, 0.25 mg/kg; 1 mL/kg) or sterile saline (N = 14; 1 mL/kg) was given to the animals immediately before they were placed onto the rotating arena without any shocks (see Section 2.3). In the acquisition phase, the 8-OH-DPAT group was randomly divided into six groups according to the treatment plan, which proceeded as follows: 30 min before being put into the apparatus, animals received subcutaneous injections of either saline (1 mL/kg), riluzole (1 mg/kg; 1 mL/kg), or memantine (1 mg/kg; 1 mL/kg). The dose of 1 mg/kg was chosen based on the results of our previous study [34], where a higher dose (5 mg/kg) had a detrimental effect on learning and locomotion in quinpirole-treated rats. Furthermore, the therapeutic effect of riluzole in the dose of 1 mg/kg has been described [35], and subcutaneous application of memantine (1 mg/kg) leads to a similar plasmatic concentration (1 μM) in rats as found in patients treated with a standard dose of 20 mg/daily [36]. Immediately before placement into the apparatus, animals received a subcutaneous injection of saline (“saline” groups) or 8-OH-DPAT (“OH” groups) at a dose of 0.25 mg/kg, which has previously been described as an effective dose [27]. The design of the treatment groups can be seen in Table 1.

### 2.3. Apparatus and Behavioral Procedures

#### 2.3.1. Active Place Avoidance Task in Carousel Maze

All experiments were conducted on a rotating arena (Carousel maze) (Figure 1A). The Carousel maze was a smooth metallic disk of a circular shape (diameter of 82 cm) enclosed by a 60 cm high transparent polyacrylic wall in a conical shape (to prevent reflections to the camera). The arena rotated clockwise (1 revolution per min) and was elevated 1 m above the floor. An unmarked to-be-avoided sector (60 degrees) was stable and defined in room-frame coordinates during the acquisition testing. In the sector, animals received a mild, 500 ms long electric shock through a subcutaneous needle implanted between their shoulders. Rats received shocks in 1800 ms intervals until they left the sector. The foot shocks were kept at the lowest possible level (0.2–0.6 mA) and were titrated for each subject, so they would be unpleasant but not painful and hence would not produce freezing in the animals (for a protocol, see Willis et al. [37]). To monitor the animals in the apparatus, they wore a small rubber jacket on their back with an infrared light-emitting diode (LED) attached. Another LED was mounted on the arena periphery and monitored the rotation, allowing the reconstructions of trajectories in the coordinate frames of the arena and room. Signals from both LEDs were captured by an analog overhead camera (Figure 1B), digitized by the DT-3155 card (Data Translation, Marlborough, MA, USA), and recorded on a PC located in an adjacent room (Figure 1C) with an online tracking program (Tracker, BioSignal Group, New York, NY, USA). An open-source software, Carousel Maze Manager [38], was used to analyze the trajectories offline and extract the evaluated parameters.

#### 2.3.2. 8-OH-DPAT Sensitization and Habituation to the Arena

The sensitization to 8-OH-DPAT was performed during the ten-session habituation to the arena (10 injections every other day), providing the chronic model. The rats were injected with 8-OH-DPAT (0.25 mg/kg) or saline (1 mL/kg). Immediately after the injection, the rats were put into the rotating Carousel maze without the activated shock sector, so they could explore freely and habituate to the rotating arena for 50 min.

#### 2.3.3. Acquisition Testing

Before the acquisition phase, conscious animals were gently implanted with a subcutaneous needle piercing the rats’ skin between their shoulders. The sharp end of the needle was cut and bent to form a small loop, which was connected to a source for the mild electric shocks via an alligator clip. In the acquisition phase, active place avoidance testing was conducted. The animals were tested every other day with a total of 10 days/10 injections. Rats were injected with riluzole, memantine (both at a dose of 1 mg/kg), or saline, 30 min before each trial. Immediately before the session, rats were injected with saline again or with 8-OH-DPAT (0.25 mg/kg, Figure 1D). Subsequently, they were placed into the maze to the side opposite the shock sector. The arena rotated for the whole 50 min of the trial at a speed of 1 revolution per minute. As the position of the “to-be-avoided” sector was not marked, animals had to use stable extra maze cues (e.g., door, shelves, windows, posters on the wall) to avoid the sector.

### 2.4. Data Analysis and Statistics

The main parameters measured were locomotion (measured as the total distance walked during the whole session (m)), the number of errors/entrances (measured as the number of entrances into the to-be-avoided sector), maximum time avoided (measuring maximum interval without entrance to the to-be-avoided sector in a given session (s)), and median speed after shock (measuring the angular velocity (deg/s)). A low number of entrances represents good learning (low errors), unimpaired memory, and an efficient avoiding strategy. Furthermore, the first and the last sessions were dissected into five consecutive 10 min intervals to assess the within-session learning (Appendix A). An independent *t*-test was used to test between-group differences on the first and the last day of habituation. Mixed-effect regression was further used to compare all treatment groups in the acquisition phase. Analysis of all parameters, with the exception of locomotion, was performed on root transformed data in SPSS (IBM SPSS, version 23, Chicago, IL, USA) and jamovi (version 1.1.9.0) software. One rat was excluded from the final analysis due to technical problems during the acquisition phase.

## 3. Results

### 3.1. Habituation

To assess whether successful sensitization to 8-OH-DPAT took place, we first compared the locomotion of rats from the 8-OH-DPAT and saline groups on the last day of habituation. Animals in the 8-OH-DPAT group had significantly higher locomotion in comparison to the saline group t(83) = −8.54, *p* < 0.001, 95% CI = [−317, 197]. Interestingly, rats sensitized with 8-OH-DPAT had significantly higher locomotion even from the first day of habituation t(789.3) = 3.44, *p* < 0.001, 95% CI = [35.48, 129.2], which shows the acute effect of the drug (Figure 2).

### 3.2. Acquisition

#### 3.2.1. Locomotion

Table 1 shows the group design. Acute treatment with 8-OH-DPAT caused significantly higher locomotion in all the “OH” groups compared to the “saline” groups that received 8-OH-DPAT only during habituation. Both riluzole and memantine in the RIL-OH and MEM-OH groups further aggravated the hyperlocomotion compared to the OH-OH group. Memantine and riluzole had, however, no effect in the “saline” groups. The “saline” groups did not significantly differ from each other

The locomotion significantly differed between groups [F(6, 90) = 35.58, *p* < 0.001], but not between sessions, as reflected by an absence of the main effect of session [F(9, 809) = 1.65, *p* = 0.098] or group*session interaction [F(54, 809) = 1.11, *p* = 0.273]. Simple and repeated planned contrasts showed that the OH-OH group had significantly higher locomotion than the OH-SAL and SAL-SAL groups (*p* < 0.001). On the other hand, the OH-SAL group did not significantly differ from the SAL-SAL group. The MEM-SAL and RIL-SAL groups also did not have different locomotion from the SAL-SAL group, the OH-SAL group, or from each other. Importantly, the MEM-OH and RIL-OH groups had significantly higher locomotion than the OH-OH group (*p* < 0.001 or *p* = 0.020, respectively), but not from one another (*p* = 0.123). For results, see Table 2 and Figure 3A.

#### 3.2.2. Entrances to the Shock Sector

The “OH” groups had a significantly higher number of entrances to the shock sector compared to the “saline” groups, suggesting decreased spatial learning ability after acute treatment with 8-OH-DPAT. Learning deficit was further exacerbated by memantine in the MEM-OH group (although not significantly) and riluzole in the RIL-OH group (significantly). There was no significant difference between the “saline” groups (Figure 3B). For a graphical illustration of typical trajectories of all treatment groups, see Figure 4.

Analysis showed significant main effects of group [F(6, 91) = 32.35, *p* < 0.001], session [F(9, 818) = 63.84, *p* < 0.001], and group*session interaction [F(54, 818) = 1.87, *p* < 0.001]. Simple and repeated planned contrasts further revealed that the OH-OH group had a significantly higher number of entrances than the OH-SAL and SAL-SAL groups (*p* < 0.001). The OH-SAL, MEM-SAL, and RIL-SAL groups did not have a significantly different number of entrances to the shock sector than the SAL-SAL group; the MEM-SAL and RIL-SAL groups also did not differ from each other or the OH-SAL group. While the MEM-OH group showed only a trend toward an increased number of entrances in comparison to the OH-OH group (*p* = 0.072), the RIL-OH group had a significantly higher number of entrances than the OH-OH group (*p* = 0.007). The MEM-OH group did not differ from the RIL-OH group in the number of entrances (for results, see Table 3). Rats had also a significantly lower number of entrances between the first and second sessions (*p* < 0.001) and the second and third sessions (*p* = 0.002). The difference between the third and fourth sessions was close to the significance level (*p* = 0.051). However, planned contrasts showed almost no significant difference in group*session interaction. The significance was probably caused by the difference between the “saline” and “OH” groups. The “saline” groups had an almost stable number of entrances from the third day on, while the number of entrances in the “OH” groups continuously decreased from the first to late acquisition sessions, although remaining very high on the last day.

#### 3.2.3. Maximum Time Avoided

Groups acutely treated with 8-OH-DPAT had a significantly lower maximum time avoided compared to the “saline” groups. Riluzole in the RIL-OH group significantly worsened the performance, leading to the RIL-OH group having a significantly lower maximum time avoided than the OH-OH group. Similarly, the MEM-OH group had a considerably, but not significantly, lower maximum time avoided compared to the OH-OH group. There was no significant difference between the “saline” groups (Figure 3C).

A main effect of group [F(6, 90) = 28.05, *p* < 0.001], session [F(9, 809) = 69.04, *p* < 0.001], and group*session interaction [F(54, 809) = 1.97, *p* < 0.001] was found. As shown by simple and repeated planned contrasts, rats in the OH-OH group were able to avoid the shock sector for a significantly shorter maximum time between two shocks than rats in the OH-SAL and SAL-SAL groups (*p* < 0.001). Again, the MEM-SAL and RIL-SAL groups did not differ from the SAL-SAL and OH-SAL groups and each other. Contrarily, the RIL-OH group had a significantly shorter maximum time avoided than the OH-OH group (*p* = 0.002), and the MEM-OH group had a substantially, although not significantly, shorter maximum time avoided than the OH-OH group (*p* = 0.054). Again, the MEM-OH and RIL-OH groups did not differ from each other (for results, see Table 4). There was also a significant difference between the first and second (*p* < 0.001), second and third (*p* = 0.008), fourth and fifth (*p* = 0.020), and the seventh and eighth sessions (*p* = 0.007), suggesting a slow increase of maximum time avoided in all groups during the whole acquisition training.

#### 3.2.4. Median Speed after Shock

Increased locomotor speed after shock, as well as visual observation of the animals, showed that all animals were able to perceive the shock and react to it. However, the “OH” groups had slower escape reactions than the “saline” groups (Figure 3D). We found significant main effects of group [F(6, 88.4) = 8.93, *p* < 0.001], session [F(9, 768) = 8.71, *p* < 0.001], and group*session interaction [F(54, 767.8) = 2.63, *p* < 0.001]. Simple and repeated planned contrasts revealed that the OH-OH group had a significantly lower median speed after shock than the OH-SAL and SAL-SAL groups (*p* < 0.001). The OH-SAL and SAL-SAL groups were not significantly different in median speed after shock, and the MEM-SAL and RIL-SAL groups did not differ from the SAL-SAL and OH-SAL groups or from each other. Similarly, the MEM-OH and RIL-OH groups were not significantly different from the OH-OH group or from each other (for results, see Table 5). Furthermore, there was a significant difference between the first and second sessions (*p* < 0.001). For significant interaction results, see interaction Appendix A.

#### 3.2.5. Entrances/Distance

The number of entrances is strongly influenced by locomotion (see Figure 3F), as hyperactive animals are more likely to enter the sector by chance. To exclude the possibility that the differences in the number of errors were only driven by locomotor activity, we calculated the ratio of the number of entrances to the walked distance for each animal and each session. Statistical analysis of this parameter showed that the “OH” groups had more entrances per unit of distance, suggesting an impairment of their ability to avoid the sector which was independent of elevated locomotion (Figure 3E).

Analysis showed a significant effect of group [F(6, 606) = 125.562, *p* < 0.001] and session [F(9, 833) = 20.729, *p* < 0.001]. Group*session interaction was not significant [F(54, 833) = 0.687, *p* = 0.958]. Simple and repeated planned contrasts revealed that the OH-OH group had a significantly higher number of entrances per walked distance compared to the OH-SAL and SAL-SAL groups (*p* < 0.001). Interestingly, the OH-SAL group was not significantly different from the RIL-SAL group, but both groups had a significantly lower entrances/distance parameter compared to the SAL-SAL (*p* < 0.001 or *p* = 0.014, respectively) and MEM-SAL groups (*p* = 0.001 or *p* = 0.030, respectively), suggesting the OH-SAL and RIL-SAL groups had a slightly lower number of entrances per similar distance than the MEM-SAL and SAL-SAL groups. Similarly, the MEM-OH group was not significantly different from the OH-OH group, but the RIL-OH had a significantly higher number of entrances per distance compared to the OH-OH (*p* < 0.001) and MEM-OH groups (*p* = 0.003) (for results, see Table 6). A significant difference was also between the first and second sessions (*p* < 0.001), and the difference between the second and third sessions was approaching significance (*p* = 0.053).

## 4. Discussion

We found that acute systemic administration of 8-OH-DPAT (0.25 mg/kg) to rats immediately before the 50 min session in the arena elicited a strong hyperlocomotion and impaired learning and avoidance of the shock sector. In addition, rats that received 8-OH-DPAT did not accelerate the escape reaction throughout training as did the control groups. Visual inspection of their reaction revealed they preserved the responsiveness to electrical shocks, but their escape route was less spatially organized to effectively leave the “to-be-avoided” sector, suggesting poor spatial knowledge of the environment. Contrary to hyperlocomotion we observed after acute administration of 8-OH-DPAT, in another study, authors described a decrease in locomotor activity and also a perseverative behavior and learning deficit [29]. However, they tested mice in a different task (non-aversive T-maze) and used higher doses of 8-OH-DPAT (1 or 2 mg/kg) than we did. It could indicate a dose-dependent effect on animal behavior with the lower doses stimulating and higher doses inhibiting impact. Alternatively, animals after 8-OH-DPAT application may react differently when solving the task under stress with an aversive motivation compared to non-aversive tasks.

We also observed only the acute effect of 8-OH-DPAT. Only the “OH” groups injected with 8-OH-DPAT (0.25 mg/kg) during habituation/sensitization and subsequently in the acquisition had higher locomotion. If the drug application was discontinued after the end of the habituation phase, sensitization to 8-OH-DPAT during habituation had no effect on spatial performance and locomotion during acquisition. Contrarily, Johnson and Szechtman [39] found that chronic administration of 8-OH-DPAT at low doses (0.0625, 0.125 mg/kg) per 8 days produced hyperlocomotion and compulsive checking even when tested after several days without the 8-OH-DPAT. However, it should be noted that the difference between spontaneous behavior (open field) and motivated behavior (carousel maze) might add up to the divergence of results. Furthermore, we tested the locomotion and cognitive skills, not the manifestation of the compulsion-like behavior per se (checking).

Memantine and riluzole significantly increased locomotion in rats acutely treated with 8-OHDPAT, even above the OH-OH group level. In the case of memantine, this effect could have been induced by the stimulatory effect of memantine itself, as it was found to produce hyperlocomotion in higher doses (5 mg/kg) [40] and dose-dependent (from 3 mg/kg) decrease of impulsivity [41]. Regarding riluzole, a previous study showed decreased behavioral and motor activity, as well as an analgesic effect, albeit at a dose four times higher (4 mg/kg) than we used in the present study [42]. At our doses (1 mg/kg), there was no effect of memantine or riluzole in the “saline” groups.

Interestingly, the learning deficit was further aggravated with the application of memantine or riluzole (1 mg/kg), and the RIL-OH group had the overall highest number of entrances to the shock sector. The RIL-OH and MEM-OH groups had only a very slight improvement over the whole acquisition testing (Figure 3B) and no improvement within a session (Appendix A). This is in agreement with the previously shown lack of any beneficial effect of memantine and riluzole on the quinpirole model of cognitive deficit related to OCD [34], but contrasts the positive memantine’s effect in relieving serotonin-induced compulsive scratching behavior in mice (however, in this case, at a ten-times higher dose, 10 mg/kg, and added to fluoxetine) [43]. In a marble-burying model of compulsive behavior, memantine (10 mg/kg) was effective in suppressing the marble-burying behavior in rats without affecting locomotion. Riluzole (10 mg/kg) was not effective in alleviating marble-burying behavior at all, although it decreased motor behavior [44].

One of the possible explanations for our observed results showing the potentiation of the 8-OH-DPAT effect with memantine and riluzole may be their action upon different brain structures. 8-OH-DPAT presynaptically blocks AMPA receptors and glutamate release through activation of 5-HT1A receptors. Nevertheless, as a 5-HT7 receptor agonist, it also enhances AMPA activity postsynaptically and CA3-CA1 synaptic transmission in the hippocampus [45]. Additionally, 8-OH-DPAT modulates glutamate transmission induced by exogenous AMPA administration [43]. Together with 8-OH-DPAT inhibiting LTP by 5-HT1A, memantine and riluzole can disturb learning and memory by further decreasing the glutamate levels. Besides the hippocampus, 8-OH-DPAT reduces excitation in the entorhinal cortex [46]. 5-HT1A and 5-HT7 receptors inhibit glutamate transmission in the frontal cortex and also in the cerebellum and many other structures involved in the motor and affective behavior (for a review, see Ciranna et al. [47]).

Our results are analogous to those of our laboratory’s previous study, which showed that riluzole and memantine exerted similar exacerbating effects in an animal model of OCD induced by dopamine receptor agonist quinpirole [34]. This shows that quinpirole and 8-OH-DPAT models might not respond well to anti-glutamatergic monotherapy by riluzole or memantine at lower doses. Possibly, higher doses of memantine or riluzole are needed to affect symptoms of quinpirole and 8-OH-DPAT animal models of OCD. However, such doses are often accompanied by side effects, such as motor inhibition or analgesia. In human studies, riluzole and memantine were effective as augmentation therapy of treatment-refractory OCD [21]. However, they did not work for all of the patients, and studies done with riluzole are limited by their small sizes. Memantine was effective in several case studies [23,24], as well as in one randomized study [25]. However, both memantine and riluzole were effective only when given together with existing treatment with SSRIs, and they were not examined alone [25,48,49]. Importantly, we measured the effect of memantine and riluzole on cognitive deficit rather than on OCD-like symptoms, which is the main focus of before mentioned human studies. Our results imply that the use of glutamate-modulating drugs in monotherapy might not be a viable treatment option for cognitive deficits induced in pharmacological animal models of OCD, although they may still work for obsessive-compulsive symptoms. It might be worth examining other doses of memantine and riluzole in future research.

## Figures and Tables

**Figure 1 biomolecules-11-01007-f001:**
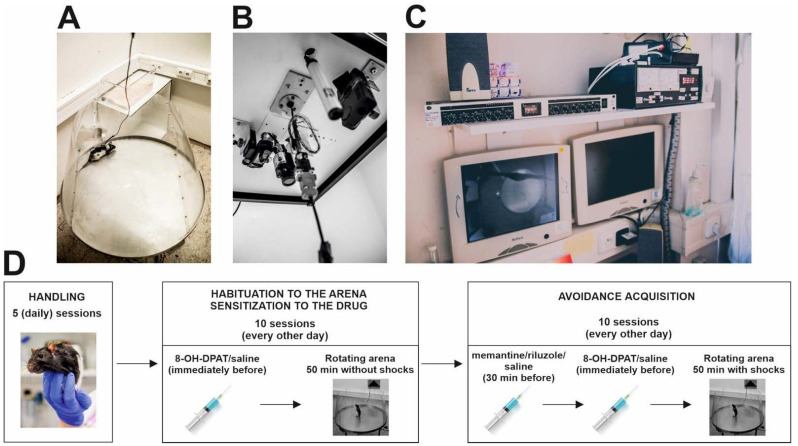
Experimental setup and design. (**A**) Rotating Carousel maze. (**B**) Camera recording the movement of experimental rats and cable with an infrared light-emitting diode attached to the ceiling above the Carousel maze. (**C**) Experiments were monitored on computers from an adjacent room. (**D**) Design of the experiment.

**Figure 2 biomolecules-11-01007-f002:**
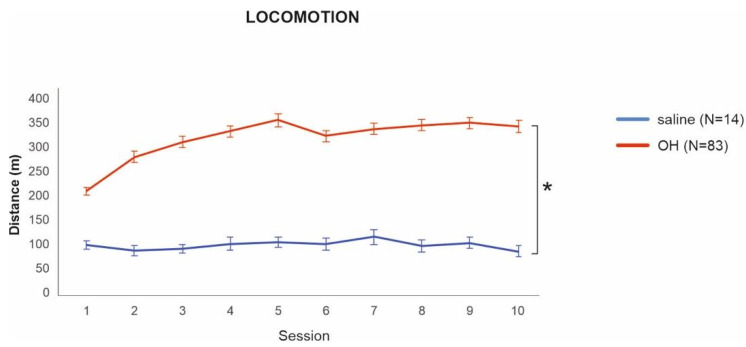
Locomotion of the 8-OH-DPAT and saline groups during habituation. Locomotion in the 8-OH-DPAT group was significantly higher from the first day of sensitization/habituation throughout all 10 sessions. * denotes a significant difference at *p* = 0.001. Data are presented as mean values ± SEM.

**Figure 3 biomolecules-11-01007-f003:**
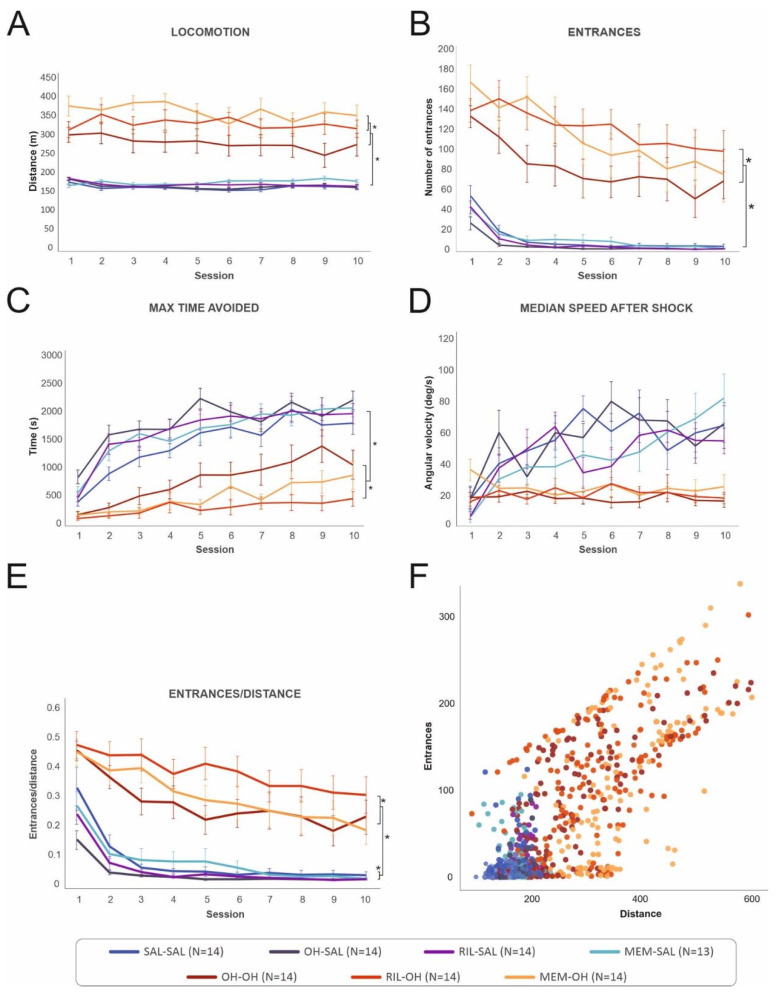
The behavior of all treatment groups during the 10 acquisition sessions in the five measured parameters. (**A**) Locomotion was stable during all 10 sessions for each group, although it was significantly higher in the “OH” compared to the “saline” groups. The MEM-OH and RIL-OH groups had significantly higher locomotion compared to the OH-OH group. (**B**) The number of entrances to the shock sector decreased across sessions, but it was significantly higher in the “OH” groups compared to the “saline” groups during all 10 sessions. The RIL-OH and MEM-OH groups had the highest number of entrances to the shock sector. (**C**) Maximum time avoided increased from the first to the last session and was significantly higher in the “saline” groups than in the “OH” groups. (**D**) Median speed after shock did not change in the ”OH” groups and only slightly increased in the “saline” groups, but with noticeable variation across sessions. (**E**) The entrances/distance parameter showed that the “OH” groups had a higher number of entrances compared to the “saline” groups, and the RIL-OH group had the highest number of entrances even when controlled for locomotion. The OH-SAL and RIL-SAL groups had the lowest number of entrances per distance. (**F**) Correlation of locomotion and number of entrances. A higher number of entrances correlated with hyperlocomotion in some animals from the “OH” groups. * denotes a significant difference at *p* = 0.05. Data are presented as mean values ± SEM with exception of Figure 3F, which presents each trial for each animal.

**Figure 4 biomolecules-11-01007-f004:**
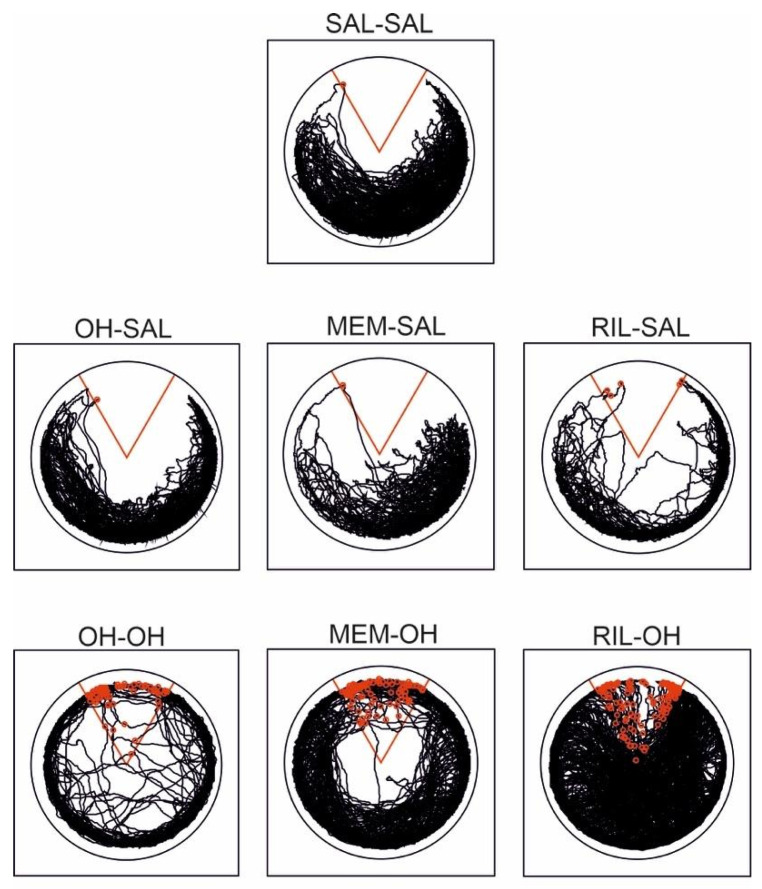
Typical trajectories of treatment groups on the 10th day of acquisition. The “to-be-avoided” sector is marked in red, and shocks received are marked as red circles. The SAL-SAL (control group) and OH-SAL groups avoided the sector well. Similarly, the MEM-SAL and RIL-SAL groups had a comparably good performance. All groups that received 8-OH-DPAT also during acquisition were more active, and they did not avoid the shock sector efficiently. The RIL-OH group had the highest locomotion and number of entrances to the to-be-avoided sector.

**Table 1 biomolecules-11-01007-t001:** The design of the treatment groups.

Habituation	N	Acquisition		N	Group	Description
		30 min before the Test	Before the Test			
Saline	14	Saline	Saline	14	SAL-SAL ^a^	Global controls
8-OH-DPAT	84	Saline	Saline	14	OH-SAL ^a^	Sensitized/undrugged/untreated
8-OH-DPAT		Memantine	Saline	14	MEM-SAL ^a^	Sensitized/undrugged/memantine treated
8-OH-DPAT		Riluzole	Saline	14	RIL-SAL ^a^	Sensitized/undrugged/riluzole treated
8-OH-DPAT		Saline	8-OH-DPAT	14	OH-OH ^b^	Sensitized/drugged/untreated
8-OH-DPAT		Memantine	8-OH-DPAT	14	MEM-OH ^b^	Sensitized/drugged/memantine treated
8-OH-DPAT		Riluzole	8-OH-DPAT	14	RIL-OH ^b^	Sensitized/drugged/riluzole treated

**^a^** All groups that received saline during acquisition are referred to as “saline” groups. **^b^** All groups that received 8-OH-DPAT during acquisition are referred to as “OH” groups.

**Table 2 biomolecules-11-01007-t002:** Simple and repeated planned contrast results of the between-group difference in the locomotion.

Parameter	Treatment Group Differences	*df*	*t*	*p*	95% CI
**Locomotion**	SAL-SAL * OH-SAL	90	0.164	0.870	−37.08, 43.860
	SAL-SAL * MEM-SAL	90	0.664	0.509	−27.27, 55.206
	SAL-SAL * RIL-SAL	90	0.368	0.714	−32.86, 48,073
	OH-SAL * MEM-SAL	90	0.503	0.616	−30.66, 51.814
	OH-SAL * RIL-SAL	90	0.204	0.839	−36.255, 44.681
	MEM-SAL * RIL-SAL	90	0.302	0.764	−32.86, 48.073
	OH-OH * SAL-SAL	90	−5.819	**<0.001**	−160.63, 79.687
	OH-OH * OH-SAL	90	−5.655	**<0.001**	−157.24, −76.295
	OH-OH * MEM-OH	90.1	3.919	**<0.001**	40.45, 121.408
	OH-OH * RIL-OH	90	2.363	**0.020**	8.33, 89.271
	MEM-OH * RIL-OH	90	−1.556	0.123	−72.61, 8.341

**Table 3 biomolecules-11-01007-t003:** Simple and repeated planned contrast results of the between-group difference in the number of entrances.

Parameter	Treatment Group Differences	*df*	*t*	*p*	95% CI
**Entrances**	SAL-SAL * OH-SAL	90	−0.839	0.404	−2.635, 1.06
	SAL-SAL * MEM-SAL	90	−0.009	0.992	−1.890, 1.87
	SAL-SAL * RIL-SAL	90	−0.413	0.681	−2.234, 1.46
	OH-SAL * MEM-SAL	90	−0.944	0.348	−2.734, 0.957
	OH-SAL * RIL-SAL	90	0.814	0.418	−1.099, 2.661
	MEM-SAL * RIL-SAL	90	0.395	0.694	−1.501, 2.260
	OH-OH * SAL-SAL	90	5.529	**<0.001**	3.360, 7.05
	OH-OH * OH-SAL	90	−6.368	**<0.001**	−7.84, 4.150
	OH-OH * MEM-OH	90	−1.82	0.072	−3.56, 0.13
	OH-OH * RIL-OH	90	2.767	**0.007**	0.76, 4.45
	MEM-OH * RIL-OH	90	0.944	0.348	−2.734, 0.957

**Table 4 biomolecules-11-01007-t004:** Simple and repeated planned contrast results of the between-group difference in the maximum time avoided of the shock sector.

Parameter	Treatment Group Differences	*df*	*t*	*p*	95% CI
**Max time avoided**	SAL-SAL * OH-SAL	90	1.858	0.067	−0.307, 11.430
	SAL-SAL * MEM-SAL	90	0.93	0.355	−3.14, 8.82
	SAL-SAL * RIL-SAL	90	1.155	0.251	−2.411, 9.326
	OH-SAL * MEM-SAL	90	−0.892	0.375	−8.702, 3.258
	OH-SAL * RIL-SAL	90	−0.703	0.484	−7.97, 3.765
	MEM-SAL * RIL-SAL	90	−0.203	0.840	−6.598, 5.362
	OH-OH * SAL-SAL	90	−3.696	**<0.001**	−995.5, −305.5
	OH-OH * OH-SAL	90	5.926	**<0.001**	11.875, 23.61
	OH-OH * MEM-OH	90	1.955	0.054	−0.0014, 11.725
	OH-OH * RIL-OH	90	−3.11	**0.002**	−15.184, −3.45
	MEM-OH * RIL-OH	90	1.155	0.251	−2.41, 9.330

**Table 5 biomolecules-11-01007-t005:** Simple and repeated planned contrast results of the between-group difference in the median speed after shock.

Parameter	Treatment Group Differences	*df*	*t*	*p*	95% CI
**Median speed after**	SAL-SAL * OH-SAL	90	0.119	0.906	−1.019, 1.150
**shock**	SAL-SAL * MEM-SAL	90	−1.3996	0.165	−1.897, 0.316
	SAL-SAL * RIL-SAL	90	−1.365	0.176	−1.838, 0.329
	OH-SAL * MEM-SAL	90	0.3999	0.689	−1.987, 3.006
	OH-SAL * RIL-SAL	90	−0.021	0.983	−2.476, 2.424
	MEM-SAL * RIL-SAL	90	−0.4204	0.674	−3.032, 1.961
	OH-OH * SAL-SAL	90	4.954	**<0.001**	1.653, 3.817
	OH-OH * OH-SAL	90	5.0849	**<0.001**	1.721, 3.880
	OH-OH * MEM-OH	90	1.089	0.279	−0.478, 1.673
	OH-OH * RIL-OH	90	0.554	0.581	−0.770, 1.378
	MEM-OH * RIL-OH	90	−0.537	0.593	−1.366, 0.778

**Table 6 biomolecules-11-01007-t006:** Simple and repeated planned contrast results of the between-group difference in the number of entrances per unit of distance.

Parameter	Treatment Group Differences	*df*	*t*	*p*	95% CI
**Entrances/distance**	SAL-SAL * OH-SAL	346.8	3.36	**<0.001**	0.034, 0.1141
	SAL-SAL * MEM-SAL	606.9	−0.4557	0.649	−0.048, 0.0299
	SAL-SAL * RIL-SAL	443.6	2.463	**0.014**	0.0105, 0.0920
	OH-SAL * MEM-SAL	550.3	3.297	**0.001**	0.0264, 0.1037
	OH-SAL * RIL-SAL	529.5	1.151	0.250	−0.0161, 0.0618
	MEM-SAL * RIL-SAL	819.5	2.1692	**0.030**	0.0041, 0.0803
	OH-OH * SAL-SAL	440.3	−9.714	**<0.001**	−0.2381, −0.1581
	OH-OH * OH-SAL	405.3	−13.552	**<0.001**	−0.312, −0.233
	OH-OH * MEM-OH	777.2	1.1267	0.260	−0.0157, 0.0581
	OH-OH * RIL-OH	782.3	4.321	**<0.001**	0.0437, 0.1164
	MEM-OH * RIL-OH	731.5	2.995	**0.003**	0.0203, 0.0974

## Data Availability

The data presented in this study are available upon request from the corresponding authors.

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
