# Peer review of "Memantine and Riluzole Exacerbate, Rather Than Ameliorate Behavioral Deficits Induced by 8-OH-DPAT Sensitization in a Spatial Task"

_biomolecules, 2021, doi:10.3390/biom11071007_

Round 1

Reviewer 1 Report

The present study by Janikova and co-workers investigated the effects of memantine and riluzole treatment in a rat model of obsessive-compulsive disorder using an active place avoidance task in the Carousel maze. They report here, that – contrarily to their expectations – memantine and riluzole treatment could not alleviate the behavioral deficits induced by 8-OH-DPAT sensitization.

Overall, this is an interesting and well-designed study, based on solid methods and the presentation of the data is clear.

I have only a few minor comments:

  • I could not find any reasoning in the text for the selection of the dosage of memantine and riluzole treatment. What was the basis of the drug dosage that they decided to use? How these dosage relates to the clinical practice?
  • For me, the most striking data was presented on Figure 3D. There, it looks like that the saline treated animals were moving in a much faster speed after the electric shock than the drug treated animals. Although apparently, this difference was not statistically significant. But based on these data, one gets an impression that the saline treated animals were more sensitive to the electric shocks. Could it be that at least some of the drug treatment effects what they observe in this test are partially due to the fact that drug treated animals are less sensitive to pain? I might be interesting to test pain sensitivity in these animals. Could the comment on these thoughts?
  • They conclude that “Our results imply that the use of glutamate-modulating drugs in monotherapy does not seem a viable treatment option for cognitive deficits induced in animal models of OCD.” I would be more careful with this conclusion. It might be that actually the behavioral test that they have used in this study is not able to detect the expected beneficial therapy-effects.

Author Response

REVIEWER 1

The present study by Janikova and co-workers investigated the effects of memantine and riluzole treatment in a rat model of obsessive-compulsive disorder using an active place avoidance task in the Carousel maze. They report here, that – contrarily to their expectations – memantine and riluzole treatment could not alleviate the behavioral deficits induced by 8-OH-DPAT sensitization.

Overall, this is an interesting and well-designed study, based on solid methods and the presentation of the data is clear.

I have only a few minor comments:

  • I could not find any reasoning in the text for the selection of the dosage of memantine and riluzole treatment. What was the basis of the drug dosage that they decided to use? How these dosage relates to the clinical practice?

Response: The dosage of memantine and riluzole (1 mg/kg) was based on results of our previous study (Janikova et al., 2019), where a higher dose (5 mg/kg) had an adverse effect on learning and locomotion in quinpirole-treated rats, mainly for the case of memantine. Also, human studies generally use memantine in the dose 10-20 mg/daily (e.g. Aboujaude et al., 2009; Bakhla et al., 2013; Gupta et al., 2019) which leads to an average plasma concentration between 0.5-1.73 μM (Parsons et al., 2007; Gomolin et al., 2017). Plasma concentration of around 1 μM has been described after a subcutaneous injection of 1mg/kg in rats (Fig. 2 in Beconi et al., 2012). Riluzole has been found effective in the dose of 1mg/kg in rats as well (e.g. Sugiyama et al., 2012).

We also inserted a short explanation of chosen dose into the text. It reads as: “The dose of 1mg/kg has been chosen based on the results of our previous study [34], where higher dose (5mg/kg) had a detrimental effect on learning and locomotion in quinpirole treated rats. Furthermore, the therapeutic effect of riluzole in the dose of 1mg/kg has been described [35] and subcutaneous application of memantine (1mg/kg) leads to a similar plasmatic concentration (1 μM) in rats as found in patients treated with a standard dose of 20mg/daily [36].”

  • For me, the most striking data was presented on Figure 3D. There, it looks like that the saline treated animals were moving in a much faster speed after the electric shock than the drug treated animals. Although apparently, this difference was not statistically significant. But based on these data, one gets an impression that the saline treated animals were more sensitive to the electric shocks. Could it be that at least some of the drug treatment effects what they observe in this test are partially due to the fact that drug treated animals are less sensitive to pain? I might be interesting to test pain sensitivity in these animals. Could the comment on these thoughts?

Response: The dose-dependent anti-nociceptive effect of 8-OH-DPAT has indeed been described (e.g. Millan et al., 1991; Bardin et al., 2001), but in higher doses than what we used in our experiment (0.25 mg/kg). A significant difference was found from a dose of 2.5 mg/kg (Millan et al., 1991) and 0.63 mg/kg twice daily (Bardin et al., 2001). Based on visual observation, we can conclude that rats that received 8-OH-DPAT before testing were able to perceive the shock, although, as you pointed out, their escape reaction was slower. Importantly, Fig. 3D also shows that all groups had the same escape reaction in the first trials and the difference is caused by an increase of escape speed in saline-treated groups, rather than a decrease of speed in 8-OH-DPAT treated groups later in the course of the experiment. Therefore it seems that the difference is caused by some other factors than decreased pain sensitivity. Of course, we cannot know what the rats feel, but we suppose that because rats in 8-OH-DPAT treated groups generally received a higher number of shocks, they might be more used to them, be less surprised after receiving one and this could lead to a slower escape speed.

They conclude that “Our results imply that the use of glutamate-modulating drugs in monotherapy does not seem a viable treatment option for cognitive deficits induced in animal models of OCD.” I would be more careful with this conclusion. It might be that actually the behavioral test that they have used in this study is not able to detect the expected beneficial therapy-effects.

Response: The sentence has been rewritten as follows: “Our results imply that the use of glutamate-modulating drugs in monotherapy might not be a viable treatment option for cognitive deficits induced in pharmacological animal models of OCD, although they may still work for obsessive and compulsive symptoms.”

Reviewer 2 Report

In this study, Janikova et al. investigated the effect of memantine and riluzole on behavioral deficits induced by chronic 8-OH-DPAT administration. On the basis of the results obtained, the authors report that both drugs exacerbated rather than ameliorated 8-OH-DPAT-induced behavioral deficits relevant for OCD. They conclude that a monotherapy with these glutamate-modulating drugs might not be an effective treatment for cognitive deficits observed in individuals with OCD. Despite these findings would be of general interest to this field of research, the work raises concern at a specific point that need to be addressed.

Major concern

  • The choice of doses in pharmacological studies is fundamental. In this study, authors do not provide any explanation about the choice of the dose of 1 mg/kg they selected for both drugs. The choice of this dose is debatable. How do they select this dose and how do they define this dose a low dose? The best approach in this kind of studies is a dose-ranging study showing the effect of different doses (from low to high) of the drugs used. Indeed, there is evidence that a lower dose of 0.2 mg/kg have pharmacological effects (PMID: 22205542). Moreover, from a translational point of view, in a previous work (PMID: 31085330) of the same group memantine has been used at a dose of 5 mg/kg, which is defined a therapeutic relevant dose. Thus, authors should perform a dose-ranging study or at least they should assess the effect of therapeutic relevant doses of both drugs.

Minor concerns

  • The rationale for testing glutamate-modulating drugs in this context is provided in the introduction but not in the abstract. This should be corrected. Authors should report the rationale of the study also in the abstract.

  • The first part of the discussion is a reiteration of the results obtained. This part should be shorten.

Author Response

REVIEWER 2

In this study, Janikova et al. investigated the effect of memantine and riluzole on behavioral deficits induced by chronic 8-OH-DPAT administration. On the basis of the results obtained, the authors report that both drugs exacerbated rather than ameliorated 8-OH-DPAT-induced behavioral deficits relevant for OCD. They conclude that a monotherapy with these glutamate-modulating drugs might not be an effective treatment for cognitive deficits observed in individuals with OCD. Despite these findings would be of general interest to this field of research, the work raises concern at a specific point that need to be addressed.

Major concern

  • The choice of doses in pharmacological studies is fundamental. In this study, authors do not provide any explanation about the choice of the dose of 1 mg/kg they selected for both drugs. The choice of this dose is debatable. How do they select this dose and how do they define this dose a low dose? The best approach in this kind of studies is a dose-ranging study showing the effect of different doses (from low to high) of the drugs used. Indeed, there is evidence that a lower dose of 0.2 mg/kg have pharmacological effects (PMID: 22205542). Moreover, from a translational point of view, in a previous work (PMID: 31085330) of the same group memantine has been used at a dose of 5 mg/kg, which is defined a therapeutic relevant dose. Thus, authors should perform a dose-ranging study or at least they should assess the effect of therapeutic relevant doses of both drugs.

Response: The dosage of memantine and riluzole (1 mg/kg) was based on results of our previous study (Janikova et al., 2019), where a higher dose (5 mg/kg) had an adverse effect on learning and locomotion in quinpirole-treated rats, mainly for the case of memantine. Also, human studies generally use memantine in dose 10-20 mg/daily (e.g. Aboujaude et al., 2009; Bakhla et al., 2013; Gupta et al., 2019) which leads to an average plasma concentration between 0.5-1.73 μM (Parsons et al., 2007; Gomolin et al., 2017). Plasma concentration of around 1 μM has been described after a subcutaneous injection of 1mg/kg in rats (Fig. 2 in Beconi et al., 2012). Riluzole has been found effective in the dose of 1mg/kg in rats as well (Sugiyama et al., 2012).

We also inserted a short explanation of chosen dose into the text. It reads as: “The dose of 1mg/kg has been chosen based on the results of our previous study [34], where higher dose (5mg/kg) had a detrimental effect on learning and locomotion in quinpirole treated rats. Furthermore, the therapeutic effect of riluzole in the dose of 1mg/kg has been described [35] and subcutaneous application of memantine (1mg/kg) leads to a similar plasmatic concentration (1 μM) in rats as found in patients treated with a standard dose of 20mg/daily [36].”

Minor concerns

The rationale for testing glutamate-modulating drugs in this context is provided in the introduction but not in the abstract. This should be corrected. Authors should report the rationale of the study also in the abstract.

Response: We included the rationale for the testing of memantine and riluzole in the abstract: “In this study, we tested spatial learning in the 8-OH-DPAT model of OCD and the effect of co-administration of memantine and riluzole - glutamate modulating agents that has been shown effective in several clinical trials.”

The first part of the discussion is a reiteration of the results obtained. This part should be shorten.

Response: We deleted few sentences from the first part of the discussion, the rest is discussed in context of other studies. We think it is necessary to provide a general summary of results in the discussion, but we shortened it to make it less repetitive.

Reviewer 3 Report

The manuscript biomolecules-1189314 by Janikova and co-workers, presents a modest study assessing the potential effects of memantine and riluzole on a spatial learning task after the administration of the agonist of the 5-HT1A and 5HT receptors 8-OH-DPAT. Given that I found a lot of critical points along all the sections of the manuscript that must be improved, I consider that the article should not be published on its present form. Below, I provide some comments that hopefully will help the authors to avoid some of these pitfalls in the future:

- The “Abstract” is plenty of abbreviations, which results in a difficult reading for the non-specialized audience of the journal (e.g., 5HT1A, 5HT7, 8-OH-DPAT).

The “N” of the subjects used in the present study is not enough relevant to mention it in the abstract section. Thence, it must be removed from the abstract.   

In line 18 of the abstract, what it means “… imperceptible to-be-avoided (shock) sector.?

In addition, it must be justified the reasoning about the use of memantine and riluzole in the present report.

Moreover, in this section, and along all the paper, they use the term “sensitization” but it is not specified for which effect is referred. That is, the behavioural sensitization must be always referred to one specific effect in response to a drug, or an exposure to a specific environment etc.

- The “Introduction” is too short. It needs to include more information for a better understanding of the rationale of the study, and to be more concise to focus the reader´s attention in the hypothesis proposed.

Moreover, as occurs in the abstract, there are a lot of abbreviations in this section.

Furthermore, there are various concepts in several sentences that are not explained and are difficult to understand for the reader. For example, in page 2 line 72, what is “…treatment-refractory OCD”?; or in line 74 of this same page, what it means “…SSRIs-refractory OCD symptoms.”?

In page 2, lines 82-84, delete the sentence “Our previous studies with a dopamine D2-like receptor agonist quinpirole showed spatial learning deficit and hyperlocomotion in sensitized rats.”. This information is not relevant for the present research and the hypothesis proposed in this study.

In line 88 of this page, it remains unknown to what drug the authors refer when they use the term “drugged”. This term is repeated across all the manuscript and it must be well described if the authors want to use it.

There is a typo in line 93: replace the symbol “(“ for “[“. In this same line of page 2, there is a reference to the protocol used in the study that must be placed in the “Materials and methods” section.

- In point “2.2. Drugs and design” of the “Material and methods” section, in line 112, page 3, the authors refers to “… section 4.3.”, but there is not a section 4.3. in this paper.

- In the description of the “Active place avoidance in Carousel maze”, in line 139 of page 4, it is necessary to specify during how time is delivered to the rats the electric shock of 0.2-0.6 mA.

- Along all the “Results” section, the nomenclature of the experimental groups in the text does not coincides with that used in the tables and figures, and this is a very confounding factor for the data interpretation to the reader. 

In addition, the results regarding the group MEM-OH X RIL-OH are not mentioned in any statistical analysis.

- In the “Discussion” section, I found several grammatical errors that must be amended.

Also, the paragraph 4 of page 12 is too much speculative in relation to the results presented in the study. My recommendation is to include some molecular experiments to support the hypothesis of the behavioural results.

Moreover, in general, the paper only presents one experiment that is analysed with different strategies but only provides one relevant result. Thus, in my opinion, the paper needs more complementary experiments with other behavioural paradigms and/or biochemical procedures to propose a strong hypothesis.   

Author Response

REVIEWER 3

The manuscript biomolecules-1189314 by Janikova and co-workers, presents a modest study assessing the potential effects of memantine and riluzole on a spatial learning task after the administration of the agonist of the 5-HT1A and 5HT receptors 8-OH-DPAT. Given that I found a lot of critical points along all the sections of the manuscript that must be improved, I consider that the article should not be published on its present form. Below, I provide some comments that hopefully will help the authors to avoid some of these pitfalls in the future:

- The “Abstract” is plenty of abbreviations, which results in a difficult reading for the non-specialized audience of the journal (e.g., 5HT1A, 5HT7, 8-OH-DPAT).

Response: Thank you for the observation. We change 5-HT to serotonin in the abstract. Unfortunately, due to the word limit of the abstract, there is not enough space to use full terms of other abbreviations, especially of 8-OH-DPAT, which is too long and might be even more confusing for the reader. Furthermore, 8-OH-DPAT is a widely used abbreviation and the drug can be found under this name on PubChem and other sites: https://pubchem.ncbi.nlm.nih.gov/compound/1220  

https://www.medchemexpress.com/8-OH-DPAT.html?src=google-product&gclid=CjwKCAjwhMmEBhBwEiwAXwFoEQKQwnHTSBcNZf_p5Uzl62arIMgtMuhaH-HwopuosE3YTh-p-w9P5RoCJpYQAvD_BwE

Another abbreviation used is OCD. OCD is defined in parenthesis in this sentence: “As such, it has been used to model obsessive-compulsive disorder (OCD)-like behavior in mice and rats.” Again, due to word limit, obsessive-compulsive disorder is then referred to as OCD.

The “N” of the subjects used in the present study is not enough relevant to mention it in the abstract section. Thence, it must be removed from the abstract.   

Response: The number of subjects has been deleted from the abstract.

In line 18 of the abstract, what it means “… imperceptible to-be-avoided (shock) sector.?

Response: The sentence has been rewritten as follows: “...where they learned to avoid visually imperceptible shock sector.” We hope it is less confusing now.

In addition, it must be justified the reasoning about the use of memantine and riluzole in the present report.

Response: The use of memantine and riluzole is justified in those sentences: “Higher glutamate concentrations were found in patients with OCD in the cerebrospinal fluid [15], the orbitofrontal cortex [16] and caudate nucleus [17], and lower in the anterior cingulate cortex [17]. Therefore, there have been attempts to study glutamate modulating agents as a possible treatment of OCD. Several clinical trials and case reports suggested the efficiency of memantine (a non-competitive low-affinity NMDA receptor antagonist) and riluzole (a drug that decreases presynaptic glutamate release by blocking sodium channels). … Riluzole is an anticonvulsant drug, nowadays officially used for the treatment of amyotrophic lateral sclerosis, and off-label it is used for the treatment of some psychiatric conditions, including OCD [21, 22]. Memantine is now used for the treatment of severe Alzheimer’s disease. Several case studies have reported its efficacy for patients with treatment-refractory OCD [23, 24]. In a meta-analysis of double-blinded, placebo-controlled, randomized studies made by Kishi et al. [25], memantine was valued as a valid treatment for patients with SSRIs-refractory OCD symptoms. Adding memantine to the SSRIs significantly improved symptoms of OCD in patients [26].”

Moreover, in this section, and along all the paper, they use the term “sensitization” but it is not specified for which effect is referred. That is, the behavioural sensitization must be always referred to one specific effect in response to a drug, or an exposure to a specific environment etc.

Response: Thank you very much for the remark, we specified the term “sensitization” in this sentence in the Introduction: “Application of this drug [meaning 8-OH-DPAT] induces behavioral sensitization, which leads to perseverative and compulsive behaviors.”   

- The “Introduction” is too short. It needs to include more information for a better understanding of the rationale of the study, and to be more concise to focus the reader´s attention in the hypothesis proposed.

Response: Thank you for the suggestion, it is true that MDPI does not specify the length of the introduction, however, generally, it is recommended to keep the introduction of research articles rather short. One paper suggests only 400 words (Araújo, 2014), another source 1.5-2 pages (https://www.elsevier.com/connect/writing-a-science-paper-some-dos-and-donts). The introduction is currently 1.5 pages long and has 803 words.

Moreover, as occurs in the abstract, there are a lot of abbreviations in this section.

Response: As in the abstract, we changed 5-HT to serotonin for a better reading flow. Other abbreviations are defined in parenthesis when mentioned for the first time. Abbreviations used in the introduction are:

  • OCD, which is defined in the first sentence of the introduction: “Obsessive-compulsive disorder (OCD) is a severe neuropsychiatric disorder affecting 1-3% of the population worldwide.”
  • SSRIs, which is defined after used for the first time in this sentence: “The hypothesis of the involvement of serotonin in the pathophysiology of OCD is popular mainly due to the effectiveness of the selective serotonin reuptake inhibitors (SSRIs) in the treatment of OCD.”
  • NMDA - the full name was missing, so we added it into the first sentence in which it was mentioned: “Drugs with different mechanisms of action have been tested, such as N-methyl-d-aspartate (NMDA) receptor antagonist amantadine …”
  • 8-OH-DPAT: the abbreviation is used for the same reason as in the abstract. It is a widely used form across other research articles.

Furthermore, there are various concepts in several sentences that are not explained and are difficult to understand for the reader. For example, in page 2 line 72, what is “…treatment-refractory OCD”?; or in line 74 of this same page, what it means “…SSRIs-refractory OCD symptoms.”?

Response: Selective serotonin reuptake inhibitors (SSRIs) are the first choice medication for obsessive-compulsive disorder, therefore SSRIs-refractory OCD is almost a synonym to treatment-refractory OCD, although treatment-refractory is a wider term, since cognitive-behavioral therapy (and its variations) are often part of the treatment and clomipramine is also regularly used when the response to SSRIs is inadequate. Patient with treatment-refractory OCD is most often defined as: “failing to achieve adequate symptom relief despite receiving an adequate course of CBT treatment and at least 2 adequate trials of SSRI medications (including clomipramine)” (Bloch & Storch, 2015).

In page 2, lines 82-84, delete the sentence “Our previous studies with a dopamine D2-like receptor agonist quinpirole showed spatial learning deficit and hyperlocomotion in sensitized rats.”. This information is not relevant for the present research and the hypothesis proposed in this study.

Response: The sentence has been deleted.

In line 88 of this page, it remains unknown to what drug the authors refer when they use the term “drugged”. This term is repeated across all the manuscript and it must be well described if the authors want to use it.

Response: The sentence has been rewritten for better understanding: “In this study, we tested these domains in chronically sensitized rats after acute administration of 8-OH-DPAT and also in chronically sensitized rats, but not after acute administration of 8-OH-DPAT.”

There is a typo in line 93: replace the symbol “(“ for “[“. In this same line of page 2, there is a reference to the protocol used in the study that must be placed in the “Materials and methods” section.

Response: The symbol has been changed and the reference transferred.

- In point “2.2. Drugs and design” of the “Material and methods” section, in line 112, page 3, the authors refers to “… section 4.3.”, but there is not a section 4.3. in this paper.

Response: Thank you very much for your observation. The numbers of sections have been corrected.

- In the description of the “Active place avoidance in Carousel maze”, in line 139 of page 4, it is necessary to specify during how time is delivered to the rats the electric shock of 0.2-0.6 mA.

Response: It has been specified: “In the sector, animals received a mild, 500 ms long electric shock through a subcutaneous needle implanted between the rat's shoulders in 1800 ms intervals until leaving the sector.” Thank you.

- Along all the “Results” section, the nomenclature of the experimental groups in the text does not coincides with that used in the tables and figures, and this is a very confounding factor for the data interpretation to the reader. 

Response: The nomenclature of experimental groups should be consistent in the text and figures, however, we sometimes refer to all groups that received saline right before the experiment in the acquisition phase as “saline” groups and to all groups that received 8-OH-DPAT in acquisition as “OH” groups, which is probably what you meant. Therefore, we explained it in the Methods section in this manner: “Immediately before placement into the apparatus, animals received a subcutaneous injection of saline (“saline” groups) or 8-OH-DPAT (“OH” groups).” And this information was also added to the footnote under table 1, which shows a design of the treatment groups.

In addition, the results regarding the group MEM-OH X RIL-OH are not mentioned in any statistical analysis.

Response: Results of MEM-OH x RIL-OH are written in tables 2-6 that show between-group differences. It was not specifically described in the text in section 3.2.1 (locomotion), but we added it into this sentence: “Importantly, the MEM-OH and the RIL-OH groups had significantly higher locomotion than the OH-OH (p < .001, or p = .020 respectively), but not from one another (p = 0.123).” In all other sections those groups are mentioned, e.g.: “The MEM-OH did not differ from the RIL-OH in the number of entrances (For results see Tab. 3).” or “Again, the MEM-OH and the RIL-OH did not differ from each other (For results see Tab. 4).”

- In the “Discussion” section, I found several grammatical errors that must be amended.

Response: Thank you, we corrected them.

Also, the paragraph 4 of page 12 is too much speculative in relation to the results presented in the study. My recommendation is to include some molecular experiments to support the hypothesis of the behavioural results.

Response: Thank you for your suggestion. Indeed, molecular experiments were not part of this completed study. We sought to describe the behavioral profile first and we believe that the results might still be of interest to other researchers in the field.

Moreover, in general, the paper only presents one experiment that is analysed with different strategies but only provides one relevant result. Thus, in my opinion, the paper needs more complementary experiments with other behavioural paradigms and/or biochemical procedures to propose a strong hypothesis.   

Response: Thank you for your remark. Other experiments were not part of the original study. Although complementary studies would be interesting, they would be performed on different groups of animals and the experiments would be quite different from what was done in this study. Consequently, some of the results would be still a bit speculative, because we would not be able to directly relate results from new groups of rats to results we already have. Therefore, it would be better, if they formed a separate article. We are convinced that this manuscript represents a complete neuropharma

Round 2

Reviewer 2 Report

The explanation provided by Janikova et al., can be only partially accepted. In Janikova et al., 2019, quinpirole was used to induce behavioral phenotypes resembling OCD. In this manuscript,  Janikova et al., used 8-OH-DPAT that may induce OCD-like behaviors through different mechanisms compared to quinpirole. Thus, it cannot be hypothesized that memantine at the dose of 5 mg/kg may worsen OCD-like behaviors because it had this effect in rats treated with quinpirole. Moreover, the same group showed that memantine, at the therapeutic dose of 5 mg/kg,  improves the performance of rats in an active place avoidance task (Wesierska et al., 2019). Thus, authors should assess the effect of memantine at the therapeutic relevant dose of 5 mg/kg.

Author Response

We agree that it could be interesting to study both lower and higher doses of memantine and we might do it in future. However, it is not possible to add the group now, as we would have to repeat the whole experiment once again with all groups to get reliable results. We chose the lower dose based on our previous research in order to reduce number of groups and focused on two different time schedules of 8-OH-DPAT application instead. As we already wrote, the 1mg/kg dose of memantine should produce plasma concentration of around 1 μM (Beconi et al., 2012), which corresponds to plasma concentration in humans who receive therapeutic doses of 10-20 mg daily (Parsons et al., 2007; Gomolin et al., 2017). Therefore the 1 mg/kg dose should have intended effect. 

We also edited conclusions to emphasize that the results are related to doses we used and avoid overgeneralization. 

Beconi MG, Howland D, Park L, Lyons K, Giuliano J, Dominguez C, Munoz-Sanjuan I, Pacifici R. Pharmacokinetics of memantine in rats and mice. PLOS Currents Huntington Disease. 2012 Feb 15 . Edition 1. doi: 10.1371/currents.RRN1291.

Gomolin IH, Papamichael MJ, Fazzari MJ, Rieger R. Memantine Plasma Concentrations Among Patients With Dementia. J Clin Psychopharmacol. 2017 Feb;37(1):117-118. doi: 10.1097/JCP.0000000000000645. PMID: 28027115.

Parsons CG, Stöffler A, Danysz W. Memantine: a NMDA receptor antagonist that improves memory by restoration of homeostasis in the glutamatergic system--too little activation is bad, too much is even worse. Neuropharmacology. 2007 Nov;53(6):699-723. doi: 10.1016/j.neuropharm.2007.07.013. Epub 2007 Aug 10. PMID: 17904591.

This manuscript is a resubmission of an earlier submission. The following is a list of the peer review reports and author responses from that submission.